

# Closed hierarchy of Heisenberg equations in integrable models with Onsager algebra

Oleg Lychkovskiy[1,2,3⋆]

**1** Skolkovo Institute of Science and Technology,
Bolshoy Boulevard 30, bld. 1, Moscow 121205, Russia
**2** Laboratory for the Physics of Complex Quantum Systems,
Moscow Institute of Physics and Technology,
Institutsky per. 9, Dolgoprudny, Moscow region, 141700, Russia
**3** Department of Mathematical Methods for Quantum Technologies,
Steklov Mathematical Institute of Russian Academy of Sciences,
8 Gubkina St., Moscow 119991, Russia

⋆ o.lychkovskiy@skoltech.ru

## Abstract

Dynamics of a quantum system can be described by coupled Heisenberg equations. In a generic many-body system these equations form an exponentially large hierarchy that is intractable without approximations. In contrast, in an integrable system a small subset of operators can be closed with respect to commutation with the Hamiltonian. As a result, the Heisenberg equations for these operators can form a smaller closed system amenable to an analytical treatment. We demonstrate that this indeed happens in a class of integrable models where the Hamiltonian is an element of the Onsager algebra. We explicitly solve the system of Heisenberg equations for operators from this algebra. Two specific models are considered as examples: the transverse field Ising model and the superintegrable chiral 3-state Potts model.

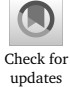

# 1   Introduction

Describing an out-of-equilibrium dynamics of a quantum many-body system is, in general, a formidable task. One can expect that this task is considerably simplified for integrable models. However, the integrability turns out to be somewhat deceptive when it comes to the dynamics. A range of methods for addressing the dynamics and resulting non-thermal steady states is under active development, including the quench action approach [1, 2], generalized Gibbs ensemble [3], generalized hydrodynamics [4,5] and advanced techniques for summing form-factor expansions [6–8]. Nevertheless, each instance of an analytical calculation in this field is a certain tour de force.

In the present paper we address the out-of-equilibrium many-body dynamics via a system of coupled Heisenberg equations. In a generic many-body model, such system of equations consists of exponentially many equations involving progressively nonlocal operators, analogously to the closely related Bogoliubov–Born–Green–Kirkwood–Yvon hierarchy of equations on reduced density matrices [9–13]. However, one can expect that in an integrable system a subset of operators can turn out to be closed with respect to the commutation with the Hamiltonian, and, as a result, the Heisenberg equations for these operators decouple from the rest of the equations. This indeed happens for a quadratic fermionic (or bosonic) Hamiltonian, where for any fixed integer $k$ a set of operators spanned by products of $k$ fermionic (or bosonic) creation and annihilation operators remains closed with respect to commutation with the Hamiltonian. This property of quadratic Hamiltonians can also be used to describe the dynamics spin-1/2 models that can be mapped to systems of noninteracting fermions. For example, in Ref. [14] an open-end transverse field Ising model was considered, and a set of Heisenberg equations for operators linear in terms of fermionic operators ($k = 1$) was derived and solved.

Here we study models with the Hamiltonian of the form

$$H = a_0 A^0 + a_1 A^1, \tag{1}$$

where $a_0$, $a_1$ are real numbers and $A_0$, $A_1$ are two operators that satisfy the Dolan-Grady conditions [15]

$$[A_0, [A_0, [A_0, A_1]]] = 16[A_0, A_1],$$
$$[A_1, [A_1, [A_1, A_0]]] = 16[A_1, A_0]. \tag{2}$$

This pair of operators generates the Onsager algebra of operators [16] (as briefly reviewed below in Sec. 2.1). This class of models includes the transverse-field Ising model [17–20] and superintegrable chiral Potts models [21]. The latter are not reduced to noninteracting fermions.

We derive and solve a system of Heisenberg equations for a set of operators $G^n, A^n$ spanning the Onsager algebra. This is done in the next section. Then, in Sec. 3, we apply the general solution to describe the out-of-equilibrium dynamics of the transverse field Ising model for a translation-invariant product initial state polarized in an arbitrary direction. In Sec. (4) we address the out-of-equilibrium dynamics of the 3-state Potts model. We conclude by the summary and outlook in Sec. 5.

## 2 Heisenberg equations and their solution

### 2.1 Onsager algebra

The Onsager algebra [16] is a Lie algebra spanned by operators $G^n$, $A^n$ that are recursively generated starting from $A^0$, $A^1$ according to

$$G^n = \frac{1}{4}[A^n, A^0], \qquad\qquad n = 0, 1, 2, \ldots,$$

$$A^{n+1} - A^{n-1} = \frac{1}{2}[G^1, A^n], \qquad\qquad n = 0, \pm 1, \pm 2, \ldots \qquad (3)$$

Commutation plays the role of the bilinear product of the algebra, with the structure explicitly given by

$$\begin{aligned}
[A^n, A^m] &= 4\, G^{n-m}, \\
[G^n, A^m] &= 2A^{m+n} - 2A^{m-n}, \\
[G^n, G^m] &= 0,
\end{aligned} \qquad (4)$$

with $G^{-n} = -G^n$. The Dolan-Grady conditions (2) are necessary and sufficient for the set of operators $A^n$, $G^n$ generated by the recursion (3) to satisfy eq. (4) [15, 22, 23].

Originally, the Onsager algebra emerged in studies of the classical Ising model [16]. An apparently first explicit construction of an Onsager algebra representation for a quantum model – namely, for the $XY$ spin chain – was presented in [24]. In an important paper by von Gehlen and Rittenberg [21] it was shown that for an arbitrary positive integer $n$ a nearest-neighbour spin-$n/2$ Hamiltonian of the form (1) can be constructed, with $A^0$ and $A^1$ satisfying the Dolan-Grady conditions (2) and thus generating a representation of the Onsager algebra. The spin-1/2 case is the transverse-field Ising model. The higher spin models are known as $n$-state superintegrable chiral Potts models or $Z_n$-invariant clock models. This discovery triggered a considerable lasting interest in such models [25–28] and, more generally, in applications of the Onsager algebra and its generalizations to quantum integrability [29–33].

### 2.2 Solving Heisenberg equations

We will work in the Heisenberg representation where a time-dependent expectation value of an observable $\mathcal{O}$ is given by

$$\langle \mathcal{O} \rangle_t = \operatorname{tr} \rho_0 \, \mathcal{O}_t, \qquad (5)$$

with $\rho_0$ being the initial state of the system and $\mathcal{O}_t$ – the Heisenberg operator related to the Schrödinger operator $\mathcal{O}$ as

$$\mathcal{O}_t = e^{iHt} \mathcal{O} e^{-iHt}. \qquad (6)$$

The Heisenberg operator satisfies the Heisenberg equation of motion,

$$\partial_t \mathcal{O}_t = i[H, \mathcal{O}_t], \qquad (7)$$

with the initial condition $\mathcal{O}_0 = \mathcal{O}$.

We are going to solve Heisenberg equations for Heisenberg operators $G_t^n$, $A_t^n$. These equations are straightforwardly obtained from eqs. (1), (4):

$$\partial_t G_t^n = 2\, i \left( a_0(-A_t^n + A_t^{-n}) + a_1(-A_t^{1+n} + A_t^{1-n}) \right),$$

$$\partial_t A_t^n = -4\, i \left( a_0\, G_t^n + a_1\, G_t^{n-1} \right), \qquad (8)$$

where $n = 0, \pm 1, \pm 2, \ldots$

We exclude $A_t^n$ from eq. (8) and obtain a system of the second-order equations:

$$\partial_t^2 G_t^n = -16\Big(a_0 a_1 G_t^{n-1} + (a_0^2 + a_1^2) G_t^n + a_0 a_1 G_t^{n+1}\Big), \qquad n = 1, 2, \ldots \qquad (9)$$

This system can be conveniently written in the compact matrix notations as

$$\partial_t^2 G_t = T G_t, \qquad (10)$$

where $G_t$ is a vector composed of $G_t^n$ and $T$ is a tridiagonal Toeplitz matrix with matrix elements $T_{nn} = -16(a_0^2 + a_1^2)$ and $T_{n\,n\pm1} = -16 a_0 a_1$.

To proceed further, we truncate the matrix $T$ to a finite $(N-1) \times (N-1)$ matrix, keeping the same notation for the truncated matrix. In fact, this truncation emerges naturally for finite-dimensional representations of Onsager algebra [26], where $A^n, G^n$ are operators on the Hilbert space of a spin chain with $N$ spins, as discussed in more detail in the next section. Anyway, the dependence on the system size goes away for local observables in the thermodynamic limit of $N \to \infty$.

We then diagonalize the truncated matrix by a unitary transformation $U$,

$$T = -U \mathcal{E}^2 U^\dagger, \qquad (11)$$

where the matrix elements of $U$ read

$$U_{mn} = \sqrt{\frac{2}{N}} \sin \frac{\pi m n}{N}, \qquad m, n = 1, \ldots, (N-1), \qquad (12)$$

and $\mathcal{E}$ is a diagonal matrix with diagonal entries

$$\varepsilon_{\varphi_n} = 4\sqrt{a_0^2 + a_1^2 + 2 a_0 a_1 \cos \varphi_n}, \quad \varphi_n = \frac{\pi n}{N}, \qquad n = 1, 2, \ldots, (N-1). \qquad (13)$$

By a standard argument, the diagonalization (11) implies the following solution of eq. (9):

$$G_t^n = \sum_{m=1}^{\infty} \Big(\partial_t c_t^{nm} G^m + 2 i c_t^{nm}\big(a_0(-A^m + A^{-m}) + a_1(-A^{1+m} + A^{1-m})\big)\Big), \qquad (14)$$

with

$$c_t^{nm} \equiv \frac{2}{\pi} \int_0^\pi d\varphi \sin(n\varphi) \sin(m\varphi) \frac{\sin(\varepsilon_\varphi t)}{\varepsilon_\varphi}, \qquad (15)$$

where the limit $N \to \infty$ is already taken. $A_t^n$ is then found from the second line of eq. (8) and reads

$$A_t^n = A^n - 4i \sum_{m=1}^{\infty} \Bigg(\Big(a_0 c_t^{nm} + a_1 c_t^{(n-1)\,m}\Big) G^m$$
$$+ 2i \Big(a_0 C_t^{nm} + a_1 C_t^{(n-1)\,m}\Big)\Big(a_0(-A^m + A^{-m}) + a_1(-A^{1+m} + A^{1-m})\Big)\Bigg), \qquad (16)$$

where

$$C_t^{nm} \equiv \int_0^t dt' c_{t'}^{nm} = \frac{2}{\pi} \int_0^\pi d\varphi \sin(n\varphi) \sin(m\varphi) \frac{1 - \cos(\varepsilon_\varphi t)}{\varepsilon_\varphi^2}. \qquad (17)$$

Eqs. (14)-(17) are the main general results of the present paper. They express time-dependent Heisenberg operators $G_t^n, A_t^n$ through the Schrodinger operators $G^n, A^n$.

# 3 Out-of-equilibrium dynamics of the Ising model

In the present section we apply our general results to the transverse field Ising model. This model belongs to the simplest class of *noninteracting* integrable spin chains since it can be mapped to noninteracting fermions [17–20]. Early studies of the transverse field Ising model mostly addressed equilibrium instantaneous and time-dependent correlation functions [14, 20, 34–40] (this topic still attracts a deal of attention [8, 41, 42]). The studies of the out-of-equilibrium dynamics, while initiated in notable early papers [34, 43], came into focus much later [8, 44–50], with most efforts directed to the dynamics after a quantum quench, as reviewed e.g. in [51]. Here we describe the out-of-equilibrium dynamics for a previously unstudied initial condition specified below.

We consider the transverse field Ising chain with $N$ spin 1/2. The Hamiltonian is given by eq. (1), where the finite-dimensional representation of the Onsager algebra reads (see e.g. [52])

$$G^n = \frac{i}{2} \sum_{j=1}^{N} \left( \sigma_j^x \left( \prod_{m=1}^{n-1} \sigma_{j+m}^z \right) \sigma_{j+n}^y + \sigma_j^y \left( \prod_{m=1}^{n-1} \sigma_{j+m}^z \right) \sigma_{j+n}^x \right),$$

$$A^n = \sum_{j=1}^{N} \sigma_j^x \left( \prod_{m=1}^{n-1} \sigma_{j+m}^z \right) \sigma_{j+n}^x,$$

$$A^{-n} = \sum_{j=1}^{N} \sigma_j^y \left( \prod_{m=1}^{n-1} \sigma_{j+m}^z \right) \sigma_{j+n}^y,$$

$$G^0 = G^N = 0, \qquad A^0 = -\sum_{j=1}^{N} \sigma_j^z, \qquad A^N = \sum_{j=1}^{N} \left( \prod_{m \neq j} \sigma_m^z \right),$$

$$G^{-n} = G^n, \qquad A^{n \pm 2N} = A^n, \qquad G^{n \pm 2N} = G^n. \tag{18}$$

Here $\sigma_j^{x,y,z}$ are Pauli matrices, $n = 1, 2, \ldots, (N-1)$, all operators $G^n, A^n$ (as well as the Hamiltonian) are assumed translation invariant with $\sigma_{N+n}^\alpha \equiv \sigma_n^\alpha$, and $\prod_{m=1}^{n-1} \sigma_{j+m}^z$ is void (i.e. replaced by 1) for $n = 1$.

Note that, thanks to the equality $G^N = 0$, the first $(N-1)$ equations in eq. (9) decouple from the subsequent equations, which rigorously justifies the truncation of the matrix $T$ in eq. (10). This also implies that our results, though presented here in the thermodynamic limit, are also valid, upon a straightforward modification, for any finite $N$.

Let us remark that the "string" operators (18) recurrently show up in the studies of the Ising model [20] and related models, the topics varying from integrable Floquet dynamics [53] to a variational ansatz for nonintegrable spin chains [54] to adiabatic gauge potentials [55].

We consider the dynamics of the Ising model prepared in a product initial state

$$\rho_0 = \prod_{m=1}^{N} \left( \frac{1}{2} (1 + \mathbf{p}\boldsymbol{\sigma}) \right), \tag{19}$$

where $\mathbf{p} = (p_x, p_y, p_z)$ is the polarization vector inside the Bloch sphere, $\mathbf{p}^2 \leq 1$. While for the initial polarisation along the $z$-axis the problem has been studied earlier [34], for a generic polarisation the dynamics has remained unexplored.

We wish to calculate time-dependent expectation values $\langle G^n \rangle_t$, $\langle A^n \rangle_t$. To this end we plug the solution (14)–(17) of the Heisenberg equations to eq. (5). The initial condition (19)

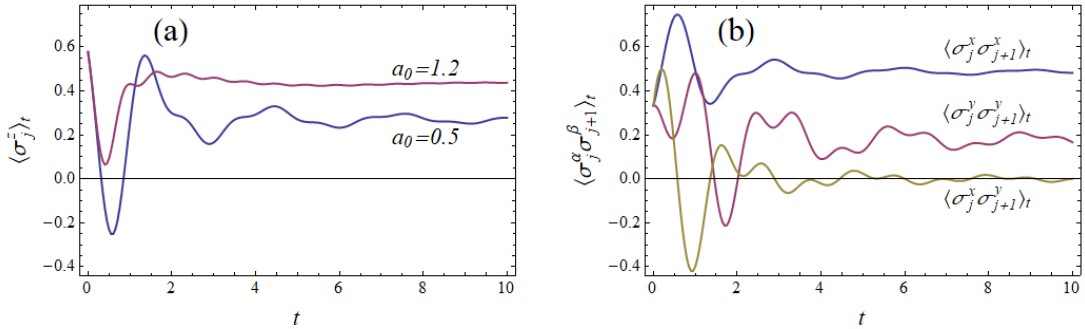

Figure 1: Time dependence of observables in the transverse field Ising model, see eqs. (21)-(23). The initial state is a product translation-invariant state (19) with the polarization $p_x = p_y = p_z = 1/\sqrt{3}$. (a) $\langle \sigma_j^z \rangle_t$ for $a_1 = -1$ and two values of $a_0$ specified in the plot. (b) $\langle \sigma_j^x \sigma_{j+1}^x \rangle_t$, $\langle \sigma_j^y \sigma_{j+1}^y \rangle_t$ and $\langle \sigma_j^x \sigma_{j+1}^y \rangle_t$ for $a_1 = -1$ and $a_0 = 0.5$.

implies

$$\langle A^n \rangle_0 = N p_x^2 p_z^{n-1}, \quad \langle A^{-n} \rangle_0 = N p_y^2 p_z^{n-1}, \quad \langle G^n \rangle_0 = i N p_x p_y p_z^{n-1}, \quad n = 1, 2, \ldots \quad (20)$$

and $\langle A^0 \rangle_0 = -N p_z$, $\langle A^N \rangle_0 = N p_z^{N-1}$. We plug these values into eq. (14) and explicitly take the sum over $m$, which is essentially a geometric series. This way we obtain in the thermodynamic limit

$$\langle A^n \rangle_t = \langle A^n \rangle_0 + 4N \int_0^\pi \frac{d\varphi}{\pi} \left( a_0 \sin(n\varphi) + a_1 \sin((n-1)\varphi) \right) \sin\varphi \left( R_\varphi \frac{\sin \varepsilon_\varphi t}{\varepsilon_\varphi} + Q_\varphi \frac{1 - \cos \varepsilon_\varphi t}{\varepsilon_\varphi^2} \right),$$

$$\langle G^n \rangle_t = iN \int_0^\pi \frac{d\varphi}{\pi} \sin(n\varphi) \sin\varphi \left( R_\varphi \cos \varepsilon_\varphi t + Q_\varphi \frac{\sin \varepsilon_\varphi t}{\varepsilon_\varphi} \right), \quad (21)$$

where

$$R_\varphi = \frac{2 p_x p_y}{1 + p_z^2 - 2 p_z \cos \varphi},$$

$$Q_\varphi = -4 a_1 \left( \frac{p_x^2 p_z - p_y^2/p_z + (a_0/a_1)(p_x^2 - p_y^2)}{1 + p_z^2 - 2 p_z \cos \varphi} + p_y^2/p_z + p_z \right). \quad (22)$$

In Fig. 1 we plot the evolution of

$$\langle \sigma_j^z \rangle_t = -N^{-1} \langle A^0 \rangle_t,$$

$$\langle \sigma_j^x \sigma_{j+1}^x \rangle_t = N^{-1} \langle A^1 \rangle_t, \quad \langle \sigma_j^y \sigma_{j+1}^y \rangle_t = N^{-1} \langle A^{-1} \rangle_t, \quad \langle \sigma_j^x \sigma_{j+1}^y \rangle_t = -iN^{-1} \langle G^1 \rangle_t. \quad (23)$$

We have verified that in a special case of $p_z = \pm 1$, $p_x = p_y = 0$ our result for $\langle \sigma_j^z \rangle_t$ coincides with that of Niemeijer [34]. Further, very recently the dynamics starting from the state with $p_x = 1$, $p_y = p_z = 0$ has been studied in detail in ref. [56]. An analytical formula for $\langle \sigma_j^x \sigma_{j+1}^x \rangle_t$ provided in [56] agrees with our result.

Let us emphasise that the results (21) have been obtained without resorting to the fermionic picture of the Ising model. Undoubtedly, the same results could be obtained in the fermionic language, since the string operators (18) are just quadratic operators in terms of fermions [24].

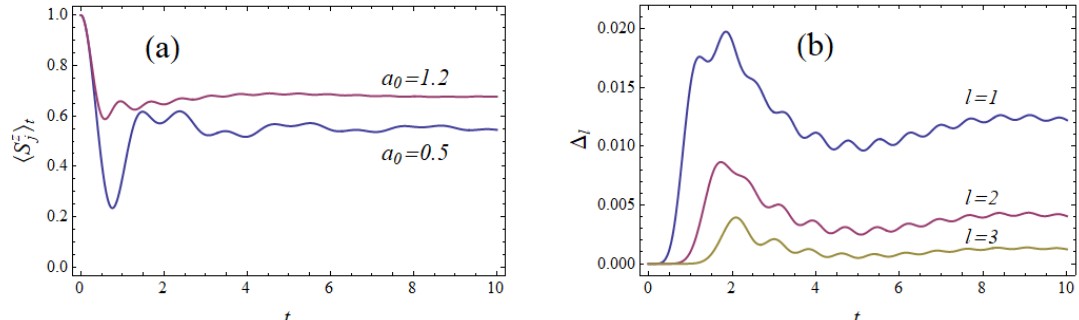

Figure 2: (a) $\langle S_j^z \rangle_t$ in the superintegrable chiral 3-state Potts model calculated via eq. (31) with the sum truncated at $l_{\max} = 4$. The initial state is the product translation-invariant state (28) with all spins polarized along the $z$-direction. The calculation is done for $a_1 = -1$ and two values of $a_0$ shown in the plot. (b) Illustration of the convergence of the series in eq. (31). Plotted is the difference $\Delta_l$ between two approximations to $\langle S_j^z \rangle_t$ derived from the truncated eq. (31), with truncation orders equal to $l_{\max} = l$ and $l_{\max} = 4$, respectively. The parameters of the Hamiltonian are $a_0 = 1.2$ and $a_1 = -1$. One can see that truncation error is small already for $l_{\max} = 1$ and rapidly vanishes with growing $l_{\max}$.

However, our method is more straightforward, both conceptually and technically. Furthermore, it avoids known difficulties of the fermionic approach related to the boundary term dependent on the overall parity $\prod_j \sigma_j^z$ [39], as well as to dealing with initial states not amenable to the generalized Wick's theorem [51].

We also note that for equilibrium time-dependent correlation functions a set of *nonlinear* equations has been derived [57, 58] and solved numerically [59]. This method relies on the Wick's theorem and, therefore, can not be generalized to the dynamics starting from non-Gaussian initial states.

## 4 Out-of-equilibrium dynamics of the 3-state Potts model

Here we consider the superintegrable chiral 3-state Potts model [21]. It is a model of $N$ spins-1 with the Hamiltonian (1), where

$$A^0 = \frac{4}{3} \sum_{j=1}^{N} \left( \frac{\tau_j}{1-\omega^*} + h.c. \right),$$

$$A^1 = \frac{4}{3} \sum_{j=1}^{N} \left( \frac{\sigma_j \sigma_{j+1}^\dagger}{1-\omega^*} + h.c. \right). \tag{24}$$

Here $\omega = e^{2\pi i/3}$ and $\tau_j$, $\sigma_j$ are operators acting on the $j$'th spin with the following properties:

$$\tau_j^3 = \mathbb{1}_j, \qquad \sigma_j^3 = \mathbb{1}_j, \qquad \tau_j^2 = \tau_j^\dagger, \qquad \sigma_j^2 = \sigma_j^\dagger, \qquad \sigma_j \tau_j = \omega \tau_j \sigma_j. \tag{25}$$

Analogously to the previous section, the system is assumed to be translation invariant with $\tau_{j+N} \equiv \tau_j$, $\sigma_{j+N} \equiv \sigma_j$. This model, in contrast to the Ising spin-1/2 chain considered above, does not map to noninteracting fermions.

We choose the following matrix representation of these operators (see e.g. [28]):

$$\tau = \begin{pmatrix} 1 & 0 & 0 \\ 0 & \omega & 0 \\ 0 & 0 & \omega^* \end{pmatrix}, \qquad \sigma = \begin{pmatrix} 0 & 1 & 0 \\ 0 & 0 & 1 \\ 1 & 0 & 0 \end{pmatrix}. \tag{26}$$

With this choice, the $z$-projection of a spin, $S^z \equiv \mathrm{diag}(1, 0, -1)$, can be expressed as $S^z = ((1 - \omega^*)^{-1} \tau + h.c.)$, and therefore $A^0$ is, up to the factor $4/3$, a total polarization along the $z$-axis:

$$A^0 = \frac{4}{3} \sum_{j=1}^{N} S_j^z. \tag{27}$$

Importantly, $\tau_j$ and $\tau_j^\dagger$ do not change the $z$-polarization of the $j$'th spin, while $\sigma_j$, $\sigma_j \tau_j$, $\sigma_j^\dagger \tau_j$ and their conjugates change the polarization by one. We will refer to the latter type of operators as *shifting* operators.

In contrast to the Ising model, an explicit closed form of $G^n$, $A^n$ is not known [28]. This means that the exact expressions (14), (16) for Heisenberg operators $G_t^n$, $A_t^n$ can not be immediately converted to exact expressions for expectation values $\langle G^n \rangle_t$, $\langle A^n \rangle_t$, as in the case of the Ising model. However, it turns out that sums in eqs. (14), (16) converge so rapidly that it suffices to calculate a few first $G^n$, $A^n$ to obtain an excellent approximation to the exact result. Below we demonstrate this by calculating $\langle S_j^z \rangle_t$ for a simple initial pure state $\rho_0 = |\Psi_0\rangle\langle\Psi_0|$ polarized along the $z$-axis,

$$\Psi_0 = |\uparrow\uparrow \dots \uparrow\rangle, \tag{28}$$

where $|\uparrow\rangle$ is the spin-up state, $S^z|\uparrow\rangle = |\uparrow\rangle$.

We employ computer algebra to calculate the first few $G^n$, $A^n$ starting from $A^0$, $A^1$ according to the recursive relations (4). The resulting expressions have a considerably more complicated appearance than those in the Ising case. For example,

$$G^1 = \frac{4}{9} \frac{1}{1 - \omega^*} \sum_{j=1}^{N} \sigma_j (\tau_j + \tau_j^\dagger - \tau_{j+1} - \tau_{j+1}^\dagger) \sigma_{j+1}^\dagger - h.c. \tag{29}$$

The number of essentially different terms necessary to represent higher $G^n$, $A^n$ rapidly grows: it equals 9 for $A^2$ and $A^{-1}$, 24 for $G^2$, 43 for $A^3$ and $A^{-2}$, 100 for $G^3$, 181 for $A^4$ and $A^{-3}$, 424 for $G^4$ *etc*. Inspecting these expressions, we conjecture the following properties of $G^n$, $A^n$.

1. Each $G^n$, $A^n$ is a sum of *strings*, where a string of length $m$ is a tensor product of $m$ operators, each chosen from the set

$$\{\tau, \tau^\dagger, \sigma, \sigma^\dagger, \sigma\tau, \tau^\dagger\sigma^\dagger, \sigma^\dagger\tau, \tau^\dagger\sigma\}, \tag{30}$$

   acting on $m$ consecutive sites.

2. Each string of length $n \geq 2$ has shifting operators at both its ends.

3. Each $G^n$ consists of strings of lengths not less than 2.

4. Each $A^n$ with odd $n$ consists of strings of lengths not less than 2.

5. Each $A^n$ with even $n$ contains strings of length 1 consisting of operators $\tau_j$, $\tau_j^\dagger$.

We leave the proof (or refutation) of these conjectures for further work. In actual calculations we use only first few $G^n$, $A^n$ where these properties have been verified explicitly.

From these properties it follows that $\langle G^n \rangle_0 = 0$ for all $n$ (in fact, this can be easily derived from the first of the recurrence relations (3)) and $\langle A^n \rangle_0 = 0$ for odd $n$. Further, for even $n$ the only type of string that contributes to $\langle A^n \rangle_0$ is the string of length 1 constructed from $\tau_j$ or $\tau_j^\dagger$. Finally, from the second line of eq. (4) with $m = 0$, one obtains $\langle A^n \rangle_0 = \langle A^{-n} \rangle_0$. As a result, we obtain from eq. (16) a simple expression

$$\langle S_j^z \rangle_t = 1 - 6\, a_1^2 \sum_{l=1}^{\infty} C_t^{1(2l-1)}\left(\langle A^{2l-2} \rangle_0 - \langle A^{2l} \rangle_0\right), \tag{31}$$

with $C_t^{1(2l-1)}$ given by eq. (17). Note that $a_0$ enters this formula through eqs. (13), (17).

In contrast to the Ising case, an explicit general formula for $\langle A^{2l} \rangle_0$ is not known. For this reason we have to truncate the sum in eq. (31) at a finite $l = l_{\max}$. Fortunately, the convergence is very rapid. The first few $\langle A^{2l} \rangle_0$ calculating iteratively from eq. (3) with the use of computer algebra read

| $l$ | 0 | 1 | 2 | 3 | 4 |
|---|---|---|---|---|---|
| $N^{-1}\langle A^{2l} \rangle_0$ | 4/3 | 4/27 | 76/729 | 1636/19683 | 12452/177147 |

We plug these values into eq.(31) and approximately compute $\langle S_j^z \rangle_t$, see Fig. 2 (a). We compare four truncations in eq. (31) with $l_{\max} = 1, 2, 3, 4$ and observe a very rapid convergence, as illustrated in Fig. 2 (b).

We conclude this section by emphasizing that a successive application of our method is conditioned to the understanding the Onsager algebra representation for a specific model under consideration. A progress in such understanding in highly desirable.

## 5 Summary and outlook

To summarize, we have solved coupled Heisenberg equations for a set of operators comprising an Onsager algebra in a system with the Hamiltonian of the form (1), which itself belongs to this algebra. The solution is given by eqs. (14)–(17). It allows one to obtain the time evolution of the corresponding observables as soon as the expectation values of these observables in the initial state are known. We have considered two specific realization of the Hamiltonian (1). The first one describes the transverse-field Ising model, where we have calculated time dependence of an infinite set of observables for a translation-invariant product initial state (19) with an arbitrary polarisation, see eqs.(21) and Fig. 1. As a second example, we have studied the superintegrable chiral 3-state Potts model. There we have obtained an approximate but highly accurate description of time evolution of the transverse polarization, see eq. (31) and Fig. 2.

Our approach can be extended to algebras different from the Onsager one. In particular, a set of $\sim 4N^2$ strictly local operators entering the sums in eq. (18) is closed with respect to commutation. Therefore, one can attempt to derive and solve Heisenberg equations for individual strictly local operators from this set. This will allow one to study the dynamics for non-translation-invariant initial states and/or transverse field Ising Hamiltonians (in particular, inhomogeneous quantum quench setups [60]) in a site-resolved manner.

An interesting open question is whether the method presented here can be extended to a broader range of integrable model. Most integrable models are not known to possess a simple algebraic structure analogous to the Onsager algebra (see, however, a recent ref. [61]

where a hidden Onsager algebra has been conjectured for the integrable XXZ spin-1/2 chain at the root-of-unity anisotropies). The absence of such structure prevents a straightforward generalization of the method.

Note, however, that one does not necessarily need an *algebra* of operators to apply the method. Indeed, the requirement of closeness of the set of operators with respect to the commutation is excessive: One actually needs a more weak property of being closed with respect to commutation with the Hamiltonian. Further work will show whether the method can be applied beyond the integrable models with the Onsager algebra.

## Acknowledgements

I am grateful to O. Gamayun, B. Fine and N. Wu for useful discussions.

**Funding information** The work was supported by the Russian Science Foundation under the grant No 17-12-01587.

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
