# Peer review of "Closed hierarchy of Heisenberg equations in integrable models with Onsager algebra"

_SciPost Physics, doi:SciPost Phys. 10, 124 (2021)_

## Round 3 · Referee Report · Anonymous (Referee 1) · 2021-4-18

Strengths

The paper present the solution of the Heisenberg equation for integrable models related to the Onsager algebra. This result allows to study Out-of-equilibrium dynamics of the Ising model and the 3-state Potts model.

Weaknesses

No discussion are given about the fact that most of the integrable models are not related to Lie algebra but to quantum groups (XXZ spin chain as exemple). In that case solving the Heisenberg equation could be more complex.

Report

I think that this that the paper is of interest and deserve to be published. If the author can comment about the case of models related to quantum groups It could be a nice addition.

---

## Round 3 · Referee Report · Anonymous (Referee 2) · 2021-4-19

Strengths

1- explicit expressions for the time-dependent expectation values of observables corresponding to operators spanning the Onsager algebra 2- exact out-of-equilibrium dynamics for the transverse field Ising model in certain translational invariant initial states of product form 3- approximate out-of-equilibrium dynamics of the polarization in the 3-state chiral Potts model for an fully polarized initial state.

Weaknesses

1- limited to Hamiltonians and observables being elements of the Onsager algebra 2- method relies on explicit representation of the Onsager algebra related to a specific model (which is known only for the Ising model) 3- site-resolution e.g. for initial states without translational invariance requires significant extension of the approach

Report

The development of methods providing analytical results for the dynamics of out-of-equilibrium many-body systems is attacting a lot of activity. In this paper the author addresses this problem by identifying cases where the Heisenberg equations of motion close to a finite set for certain classes of observables (as in models of non-interacting fermions or bosons). Specifically, he considers (superintegrable) models where the Hamiltonian is an element of the Onsager algebra.

In this case, the time evolution of the operators $A^n$, $G^n$ spanning this algebra is given by a system of linear equations of motion. Moreover, representations of the Onsager algebra related to spin chains of given length are finite dimensional [26] which allows for a truncation of the system to a finite one. These equations are solved giving explicit expressions for the time-dependent (Heisenberg) operators spanning the Onsager algebra for which the thermodynamic limit can be taken.

The author applies this general result to two specific models. For the transverse field Ising model closed forms for the $A^n$, $G^n$ are known. For a class of translational invariant initial states of product form this allows to obtain compact expressions for the time-dependent expectations values for these operators, extending previous results obtained in the fermionic formulation of the Ising model [34].

The second example considered in the manuscript is the 3-state chiral Potts model. Here the representation of the $A^n$, $G^n$ in terms of local spin operators is not known for general $n$. The author observes, however, that taking into account the first few of these operators into account leads to rapidly converging approximations to the exact expectation value. This is demonstrated for the time dependent local polarization in a completely polarized pure initial state.

In summary, the use of the Onsager algebra provides a straightforward way to study the dynamics of the corresponding superintegrable models with sufficiently simple, in particular translationally invariant initial states.
Adding strictly local operators to the Onsager algebra may well allow to address problems without translational invariance thereby complementing existing approaches. This appears feasible but will require a significant extension of the approach.

I recommend publication of this manuscript in SciPost Physics.

---

## Round 4 · Referee Report · Anonymous (Referee 1) · 2021-5-2

Report

The author's could also look to
T. Deguchi, K. Fabricius and B.M. McCoy, J. Stat. Phys. 102, 701 (2001)
about loop algebra symmetry at root's of unit.
  • validity: -
  • significance: -
  • originality: -
  • clarity: -
  • formatting: -
  • grammar: -

Author:  Oleg Lychkovskiy  on 2021-05-06  [id 1413]

(in reply to Report 1 on 2021-05-02)
Category:
pointer to related literature

I thank Referee for bringing this reference to my attention. It would be interesting to try to extend the approach developed in my paper by exploiting the said algebraic structure.

---

## Round 4 · Referee Report · Anonymous (Referee 2) · 2021-5-3

Report

The author has added a remark on the possible extension to integrable models without an algebraic structure similar to the one considered in the manuscript and added a reference to a conjectured 'hidden' Onsager algebra structure in the XXZ spin chain at certain values of the anisotropy.

I recommend to publish the paper in the present version in SciPost Physics.

---

## Round 4 · Author Response

I thank Editor and Referees for their time and effort to review my manuscript, and for recommending it for publication.

The second Referee and the Editor have requested to comment upon the applicability of the method to a wider range of integrable models. I agree with the second Referee that most integrable models are not known to have a simple underlying algebraic structure similar to the Onsager algebra. A straightforward generalization of our method to such models is therefore unlikely. I would like to point out, however, to a very recent ref. [61], where a hidden Onsager algebra has been conjectured for the integrable XXZ spin-1/2 chain at the root-of-unity anisotropies. I have added these considerations to the "Summary and outlook" section. I believe further work could elucidate the scope of the method.

---

## Round 4 · List of Changes

1) Following the advise by the second Referee, I have added the following paragraph to the "Summary and outlook" section:

An interesting open question is whether the method presented here can be extended to a broader range of integrable model. Most integrable models are not known to possess a simple algebraic structure analogous to the Onsager algebra (see, however, a recent ref. [61] where a hidden Onsager algebra has been conjectured for the integrable XXZ spin-1/2 chain at the root-of-unity anisotropies). The absence of such structure prevents a straightforward generalization of the method.

2) I have mentioned that my result in a particular case agrees with a very recent preprint [56], see the text below eq. (23).

3) I have added a line below eq. (31) explaining how this equation depends on a_0.

4) Several references have been updated.

---

## Editorial Decision

published